# Subsistence strategy mediates ecological drivers of human violence

**Weston C. McCool**[1,2☯]*, **Kenneth B. Vernon**[1,2,3☯], **Peter M. Yaworsky**[2,4☯], **Brian F. Codding**[1,2,3☯]

**1** Department of Anthropology, University of Utah, Salt Lake City, UT, United States of America, **2** Archaeological Center, University of Utah, Salt Lake City, UT, United States of America, **3** Global Change and Sustainability Center, Salt Lake City, UT, United States of America, **4** Department of Archaeology and Heritage Studies, Aarhus University, Aarhus, Denmark

☯ These authors contributed equally to this work.
* weston.mccool@anthro.utah.edu

**Data Availability Statement:** All of the data presented in this paper can be found in the Supporting Information.

**Funding:** The authors received no specific funding for this work.

## Abstract

Inter-personal violence (whether intra- or inter-group) is a pervasive yet highly variable human behavior. Evolutionary anthropologists suggest that the abundance and distribution of resources play an important role in influencing differences in rates of violence, with implications for how resource conditions structure adaptive payoffs. Here, we assess whether differences in large-scale ecological conditions explain variability in levels of inter-personal human violence. Model results reveal a significant relationship between resource conditions and violence that is mediated by subsistence economy. Specifically, we find that interpersonal violence is highest: (1) among foragers and mixed forager/farmers (horticulturalists) in productive, homogeneous environments, and (2) among agriculturalists in unproductive, heterogeneous environments. We argue that the trend reversal between foragers and agriculturalists represents differing competitive pathways to enhanced reproductive success. These alternative pathways may be driven by features of subsistence (i.e., surplus, storage, mobility, privatization), in which foragers use violence to directly acquire fitness-linked social payoffs (i.e., status, mating opportunities, alliances), and agriculturalists use violence to acquire material resources that can be transformed into social payoffs. We suggest that as societies transition from immediate return economies (e.g., foragers) to delayed return economies (e.g., agriculturalists) material resources become an increasingly important adaptive payoff for inter-personal, especially inter-group, violence.

## Introduction

Inter-personal violence, whether intra- or inter-group, is a persistent attribute of human societies, though the degree of violence varies [1]. Explanations for coalitional violence include a wide variety of cultural [2–5] and evolutionary [6–17] hypotheses. These long-debated explanations center on the ultimate causes of collective violence while evolutionary approaches focus on the adaptive payoffs for individual participation. Recent research also explores the co-evolution of inter-group violence with group cooperation, altruism, and the emergence of

**Competing interests:** The authors have declared that no competing interests exist.

complex social systems [16, 18, 19]. Those that propose inter-group violence to be a rare or maladaptive behavior [20–22] have been effectively countered in many cases [23–26], although important debate persists [7].

Here we examine how ecological and economic factors influence the adaptive payoffs for inter-personal violence and in doing so explain cross-cultural variation. We focus on the proposed adaptive payoffs for participation in inter-group violence–although these payoffs should impact intra-group violence as well–which are most often posited to be (a) social rewards such as status, mating opportunities, and alliance formation [8, 11, 13, 27], or (b) the procurement and defense of scarce resources [6, 28–30]. Our central question is whether macroscale variation in the abundance and distribution of local resources has a structuring effect on the adaptive payoffs for violence, and whether subsistence economy mediates this relationship. It is critical then that we introduce how social and material payoffs relate to violent interactions.

Many scholars hypothesize that inter-group violence serves as a resource procurement strategy and, thus, propose that the benefits of violence are high when individuals compete for high-ranking resources that are scarce and distributed in patches that can be effectively privatized [6, 28–30]. In the most general sense, inter-group violence is proposed to be a subsistence activity to procure or defend valued resources. Typically, these strategies target group-level rewards, such as territory, that elicit collective action problems [11, 31, 32]. As such, participation in resource violence must be accompanied by social rewards, such as status [11], or a disproportionate share of the loot that can then be mobilized to enhance fitness. Scarce, patchily distributed resources, thus, provide the initial incentive for violence, with social rewards or the accumulation of personal wealth providing the impetus for individual participation.

Other researchers propose that participation in inter-group violence is best explained as a strategy to directly gain, co-opt, or defend status, alliances, or mating opportunities (i.e., social rewards), with resource procurement and defense playing an irrelevant or epiphenomenal role [11, 13, 27, 33–35]. This may arise when socioeconomic conditions (e.g., mobility, resource surplus, privatization) preclude the use of violence to amass and own resources. In this sense, participation in violence is motivated by direct social rewards and should be more frequent when the abundance of local resources allows for energetic surpluses to be allocated into violent competition [30, 32, 36–38]. In this case, rich environments more frequently "finance" costly inter-group violence.

Relating these alternative explanations to resource conditions (Fig 1), we expect that (1) if inter-group violence is incentivized by the procurement of material resources and attendant fitness payoffs, then rates of violence should be highest in marginal, heterogeneous environments with scarce, clustered resources that can be monopolized [6, 28–30]. Alternatively, (2) if inter-group violence is primarily a strategy to directly acquire social rewards, with resource acquisition being an epiphenomenal motivation [11, 33, 34], then rates of violence should be highest in environments where abundant and predictable resources provide energy surpluses that can be diverted into violent inter-group (and likely intra-group) competition [30, 32, 36–38]. A third, somewhat less obvious possibility is that subsistence economy plays a key mediating role, whereby the transition to delayed-return economies (e.g., agriculturalists) allows for the accumulation of wealth, generating greater incentives for resource violence.

We evaluate the importance of resource conditions relative to levels of violence using a global archaeological dataset of delayed-return agricultural societies and a global ethnographic dataset of immediate-return forager and mixed-economy horticultural societies (Fig 2). We combine archaeological and ethnographic datasets because (a) archaeological violence datasets are extremely limited for foragers and horticulturalists, while ethnographic research lacks data on violence among complex, agricultural societies–particularly non-industrial and non-western populations prior to the 20[th] century, and (b) it is vital that we assess whether

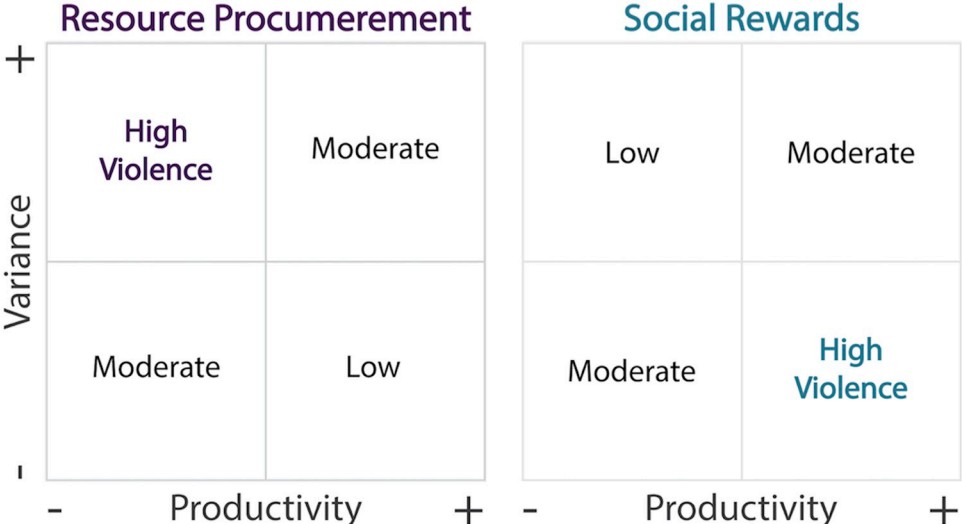

**Fig 1. Conceptual model.** Expectations for the two primary explanations. Punnett squares display the expected payoffs for violence as a function of environmental productivity (resource abundance) and heterogeneity (resource dispersion).

socioeconomic features mediate the ecological drivers of violence, as discussed above. We combine these datasets to test how levels of violence are affected by the spatial distribution of net primary productivity (NPP), both its mean (resource abundance) and standard deviation (resource dispersion). The novel dataset presented here expands on previous evaluations of coupled archaeological and ethnographic data [e.g., 18] and allows us to quantitatively explore

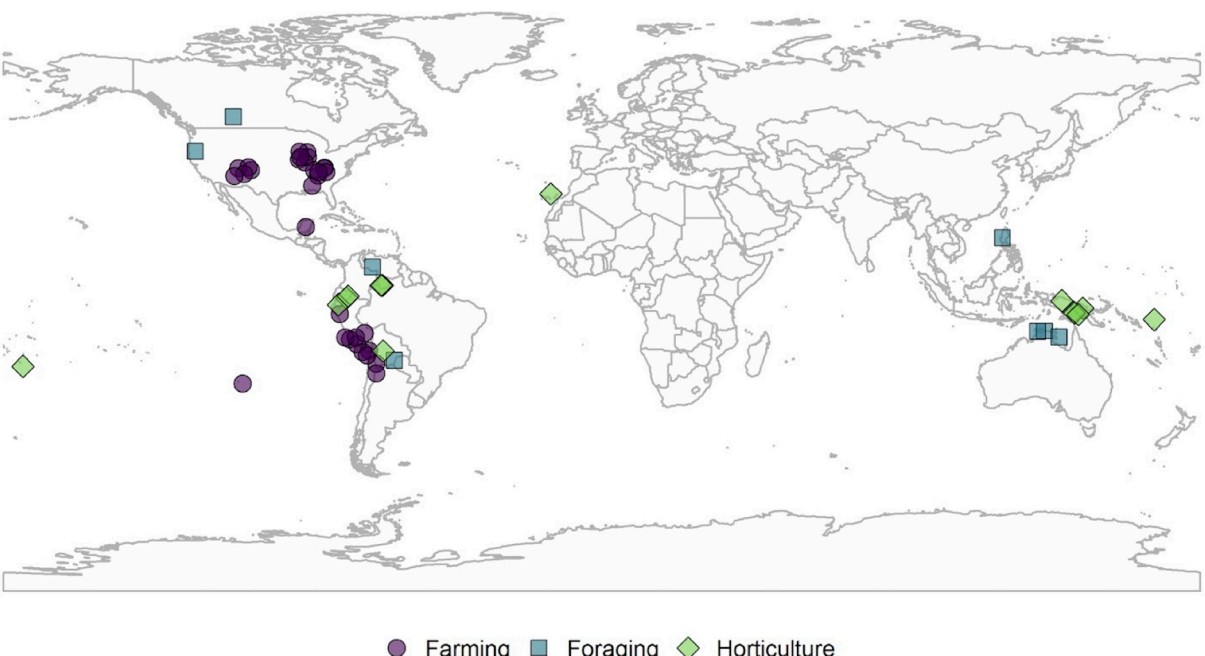

**Fig 2. Map of observations.** The global distribution of societies with violence data used in this study; color and shape coded by mode of subsistence.

the question of whether levels of violence differ by ecological conditions and modes of subsistence.

## Materials and methods

Following Bowles [18], we developed a database of proportional violence (hereafter referred to as "PV") using archaeological and ethnographic data and categorized all sampled groups into three modes of subsistence (MoS): foraging, horticulture, and farming (see, S1 File). We define MoS simply as benchmarks along the continuum from immediate-return to delayed-return economies along with the related features of subsistence and social organization such as mobility, surplus, storage, and privatization [39, 40]. Pastoralists are not included in this study due to the lack of sufficient observations to run a separate empirical model. In all, the database contains 53 populations from seven World regions (Fig 2 and S1 and S2 Files).

While archaeological and ethnographic measures of violence are not identical, they have been used to good effect in several prominent studies [18, 41]. The dataset developed here builds off prior studies by providing additional archaeological and ethnographic samples and improves precision by excluding archaeological groups with small sample sizes.

### Archaeological measure of violence

The archaeological database derives from several dozen peer-reviewed publications that report proportions of violent skeletal trauma. Bioarchaeologists have robust methods for distinguishing intentional violent trauma from accidents and post-mortem damage [42–44]. This database operationalizes PV by defining it as the number of individual skeletons with evidence of violent trauma divided by the total skeletal sample for each archaeological observation (i.e., region or society). Published data that reports ante-mortem (healed) and peri-mortem (non-healed) skeletal trauma were included if they (1) reported proportion of violent trauma that exclude accidents and post-mortem damage, (2) contained sample sizes of approximately n = 100 individuals or greater (with the exception that small islands contain smaller samples sizes), and (3) contained site or region-specific location data. In all, 34 samples from seven World regions were included from 25 sources (S1 and S2 Files). Since most sources do not report peri-mortem trauma rates, we rely on generalized inter-personal violence. Because this type of data does not distinguish intra- vs inter-group violence, we treat the database as a long-term record of generalized inter-personal violence for a given area, which is standard practice for cross-regional bioarchaeological studies [e.g., 6, 42, 45–48]. However, as intra-group lethal and non-lethal violence tend to be low in human societies [1, 2], we expect that changes in population-level inter-personal violence will largely reflect fluctuations in inter-group violence. Furthermore, the payoffs for violence outlined in this paper (e.g., mating, status, and resource competition) should influence general inter-personal violence, regardless of whether it is intra- or inter-group. While certain global regions are not included due to lack of data with adequate sample sizes, the current samples in the archaeological database span a variety of net primary productivity (NPP) values (Fig 3), which allows for robust hypothesis testing. To provide a more representative sample of archaeological data, research will be needed to produce and disseminate high-resolution trauma datasets from a wider variety of economic and ecological contexts.

### Ethnographic measure of violence

Our ethnographic database derives from peer-review publications and online scholarly sources. This database operationalizes PV by defining it as the number of observed or reported acts of lethal violence divided by the estimate of the total ethnographic population size.

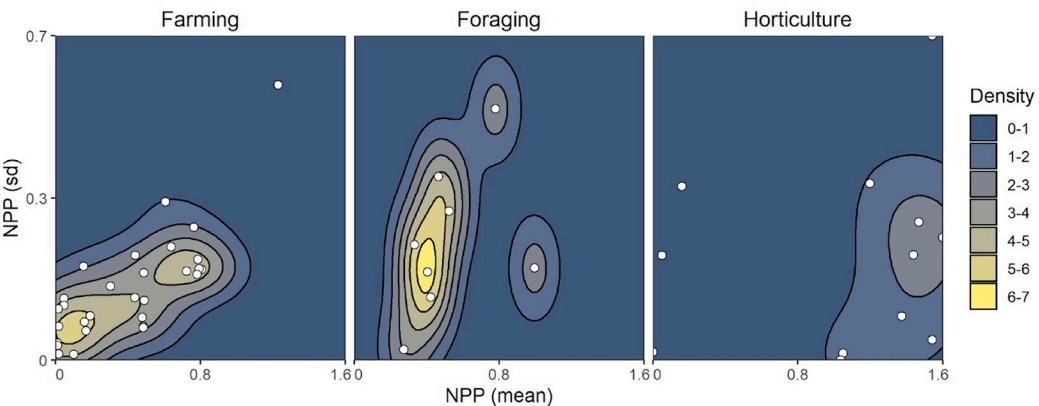

**Fig 3. A density plot for the observations (points) in our dataset.** Color represents the density of observations (societies) along both environmental dimensions (mean NPP and standard deviation in NPP).

Proportions of lethal violence are obtained first through aggregate sources (e.g., ourworldindata.org) and then corroborated using eight original published sources (SS2 File). In all cases, proportions of violent death are obtained from the most recent publications. The ethnographic database contains 19 populations from seven World regions (Fig 2).

## Data compatibility

Because the archaeological database contains generalized inter-personal trauma and the ethnographic database contains violent deaths, these two databases are not perfectly compatible. Perhaps most importantly, we cannot compare these two datasets to evaluate differences in absolute rates of violence. We must acknowledge that the ethnographic rates would be higher if they included all skeletal trauma, and the archaeological rates would be lower if they included only lethal trauma. That said, both data types produce similar types of measurements: incidents of inter-personal violence divided by population estimates. In addition, lethal and sub-lethal violence tend to co-occur, as does intra- and inter-group violence [49–52] and should offer comparable measures of *relative* rates of violence in each society, thus informing us about the functional relationship between that relativized rate and the environment.

What does this mean for our models? Primarily, it means that they cannot provide an unbiased estimate of the intercept, understood as the marginal or baseline rate of violence. However, they should offer unbiased estimates of the coefficients, where those capture functional responses to environmental conditions. Fortunately, for this analysis, the intercepts are largely irrelevant, for all we need to evaluate our hypotheses are the model coefficients. That is, we are more concerned with relative rather than absolute rates of violence.

## Environmental productivity

As a proxy for environmental productivity, we relied on terrestrial net primary productivity (NPP), which approximates photosynthesis, measuring the amount of energy that is turned into mass and thereby approximating the amount of new growth biomass available to consumers. The rasters containing mean and SD NPP were compiled between 2000 to 2015 at a 1-km resolution using remotely sensed data from the MODIS instrumentation on NASA's Terra satellite, processed and provided by the Numerical Terradynamics Simulation Group at the University of Montana [53, 54]. These were converted to kg/C/m2 by multiplying by the scaling factor 0.0001 [55]. We then extract NPP estimates for a 50km buffer generated around a

geospatial centroid for each of the societies in our sample and map out the densities of the observations over space (Fig 3).

Modern NPP has been used to predict numerous prehistoric phenomena including population density [56–58], habitat colonization [59], resource scarcity [6] and more. While NPP provides a measure of modern productivity, the *relative* NPP ranking of each region should have remained consistent over time due to their broad geographic range representing general physiographic regions rather than variants within single ecosystems [60, 61]. To better illustrate this point, we take as an example two regions included in our sample: the North American Colorado Plateau (CP) and the Illinois river valley (IRV). While the absolute NPP of these two regions certainly fluctuated throughout the centuries, there was no point during which the "mountain grassland and scrublands" of the CP were more environmentally productive (higher NPP) than the "tropical and subtropical dry broadleaf forests" of the IRV [60, 61]. The *relative* differences in environmental productivity between these two regions are due to their differing physiographic characteristics resulting in a marginal ecology with the former and a productive ecology with the latter, which have remained comparatively consistent for millennia. The same can be said about the relative variation in NPP between the other regions compared in our analysis and the observations clustered therein (e.g., the lowland neotropics of the Amazon Basin versus the Canadian boreal forests). At most, changes in absolute NPP would only serve to exaggerate or flatten the modeled curves but would not lead to trend reversals.

## Analytical methods

We first evaluate our sample for potential spatial autocorrelation in proportion of violence (sigma) using Monte Carlo simulations of a Moran's I Test as implemented by the *moran.mc* function in the R *spdep* package [62] (SS2 File). This includes an analysis across all MoS and within each MoS. We find the former to be significant (MI = 0.37, $p$ = 0.03) but primarily driven by autocorrelation among farming populations (MI = 0.39, $p$ = 0.045), as there was no autocorrelation among foragers ($p$ = 0.663) or horticulturalists ($p$ = 0.295). To test whether this may indicate a latent spatial process an additional Moran's I Test on the residuals of the full model (see below) revealed no significant spatial autocorrelation (MI = 0.169, $p$ = 0.165), suggesting that model coefficients captured any latent spatial process in the response.

To test our research questions, we fit three generalized linear models (GLMs) with a binomial distribution and log link appropriate to proportional data using quasi-likelihood estimation to account for overdispersion in our response variable (PV). We generate a null model without predictors, a base model with only our environmental predictor variables (mean and SD NPP), and a complete explanatory model with environmental predictors and MoS, specified as an interaction term to test for differences in intercepts and slopes, with farming as the reference class. We then perform a likelihood ratio test to evaluate whether each increase in model complexity provides sufficient gain in explanatory power. For each model, we report PV as a function of mean and SD NPP along with coefficient estimates and standard errors shown as log of the odds ratio. All analyses are conducted in the R programming environment and language [63] (For more details about our analysis, please see SS2 File).

## Results

The base empirical model that includes all forager, farmer, and horticulturalist populations results in a significant model improvement from the null model ($X^2$ = 0.47, $p$ = 0.0458, Table 1) and shows that violence positively covaries with mean NPP ($\beta$ = 0.513, $p$ = 0.0475), and has a limited response to NPP SD ($\beta$ = -1.355, $p$ = 0.1771), suggesting that the overall

**Table 1. Results of binomial GLMs evaluating proportional violence (PV) as a function of mean and SD NPP.** Coefficient estimates and standard errors are shown as log of the odds ratio. Coefficients in the full model are relative to the farming reference class.

| Covariate | Coefficient | Std. Error | P-Value |
|---|---|---|---|
| **Null Model** | -1.5041 | 0.1202 | <0.0001 |
| **Base Model** | | | |
| Intercept | -1.5696 | 0.2239 | <0.0001 |
| NPP | 0.5134 | 0.2526 | 0.0475 |
| NPP SD | -1.3545 | 0.9895 | 0.1771 |
| **Full Model** | | | |
| Intercept | | | |
| Farming | -1.7953 | 0.2448 | <0.0001 |
| Foraging | -0.0017 | 0.7239 | 0.9981 |
| Horticulture | 0.1434 | 0.5506 | 0.7957 |
| NPP | | | |
| Farming | -2.782 | 0.737 | 0.0005 |
| Foraging | 5.251 | 1.285 | 0.0002 |
| Horticulture | 3.546 | 0.826 | 0.0010 |
| NPP SD | | | |
| Farming | 6.956 | 2.114 | 0.0020 |
| Foraging | -11.266 | 3.030 | 0.0006 |
| Horticulture | -8.785 | 2.402 | 0.0007 |

proportion of violence is slightly higher in productive environments ($r^2 = 0.08$), as predicted by the social rewards hypothesis (Table 1).

The more complex empirical model accounting for MoS (forager, horticulture, farmer) results in a significant model improvement from the base model ($X^2 = 2.13$, $p < 0.0001$, $r^2 = 0.34$, Fig 4). The proportion of violence as a function of environmental productivity significantly differs between foragers and farmers ($p = 0.0002$), between farmers and horticulturalists ($p < 0.0001$), and between foragers and horticulturalists ($p = 0.0001$) (Table 1 and Fig 4). The proportion of violence as a function of environmental heterogeneity also differs significantly between these MoS. For foragers and horticulturalists, violence positively co-varies with

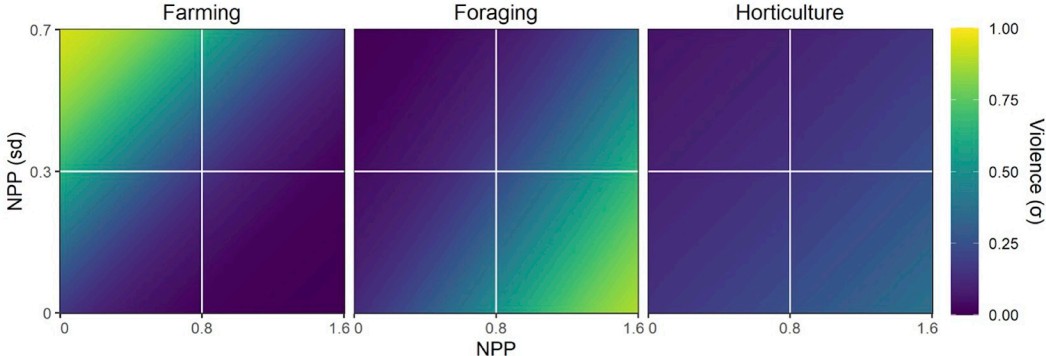

**Fig 4. Model results showing the predicted response of proportional violence (PV) for each subsistence mode to every combination of mean environmental productivity (NPP) and the standard deviation in environmental productivity (NPP SD).** Predicted values are constrained to their observed range in the data. Marginal response plots for each MoS are available in S1 File.

environmental productivity and homogeneity, while for farmers violence negatively co-vary with productivity and homogeneity.

To summarize, including MoS as an interaction term significantly improves model fit, showing that proportions of generalized violence among foragers and farmers in particular exhibit opposite covariance with environmental productivity and heterogeneity.

## Discussion

The empirical model shows that farmers have higher proportions of violence in low productivity, high heterogeneity environments and the lowest proportions of violence in high productivity, homogeneous environments. The opposite trend is present among foragers (and to a lesser extent, horticulturalists), with the highest proportions of violence occurring in high productivity, homogeneous environments and the lowest proportions of violence occurring in low productivity, heterogeneous environments. These results show that in order to explain ecologically driven variation in human violence it is necessary to account for different subsistence economies. The opposite covariance along subsistence lines warrants explanation in relation to the adaptive motivations for violence.

These results support the expectation that the adaptive payoffs for inter-group (and interpersonal) violence are increasingly mediated by resource availability as societies develop delayed-return economies reliant on resource surpluses, storage, and privatization, all of which enable the accumulation and distribution of wealth that can be mobilized to enhance or maintain fitness rewards such as status, marriage opportunities, and alliances [64–66]. For immediate-return economies, like mobile foraging, the benefits of violence are high when resources are abundant, homogeneous, and predictable, as their economic and social organization renders resource procurement an epiphenomenal payoff relative to direct social rewards. The result of this transition is a fundamental change in the socioecological conditions that promote collective violence. While our explanation is tentative and will require further testing, it provides a single evolutionary rationalization for these seemingly divergent findings. We further unpack this explanation below.

### Violence over social and material rewards

Violence used to gain, co-opt, or defend direct mating and marriage opportunities has long been a topic of controversy and empirical debate [e.g., 67]. Ethnographers report evidence showing adult males frequently fight over mating opportunities and real or perceived infidelities [13, 33, 67]. Others find that violent conflict can result in the disproportionate accumulation of wealth that can be used to gain direct mating opportunities via social systems such as bride price [12]. In either case, those who hypothesize direct mating opportunities as a prime motivation for violence emphasize its potential to directly elevate inclusive fitness [8, 12, 13, 33]. However, explanations that promote mating opportunities without reference to status competition are problematic, as reputation is tied up in all competitive activities, including mating contests.

Afterall, competition takes numerous forms, including behavioral strategies aimed at signaling fitness quality to potential mates, allies, and competitors [68–71]. Many scholars provide evidence that participation in coalitional violence acts as a form of costly signaling intended to deter competitors, encourage alliance formation, and attract mates [8, 11, 28, 35]. Observers of these costly displays benefit from honest signals that are useful for conflict avoidance, mate choice, and competitor evaluation. As the costs of participating in coalitional violence tend to be high [6, 72], dishonest signals should be rare and easily detectable. These payoffs have clear fitness consequences, as status consistently links to reproductive success

[73]. Violence as a costly display also includes fights over direct mating opportunities, which signal one's willingness to defend or co-opt access to reproductive partners. To summarize, we argue that the available evidence points to status competition as a key adaptive payoff for human violence, with conflict over mating opportunities being a component of larger reputational contests. Environmental conditions then must be linked to the variable payoffs for violent status contests, and how they are mediated by subsistence economy.

For foragers and horticulturalists, organization or participation in collective violence can produce or signal embodied and relational wealth [74] that may translate into reproductive success [73]. For agriculturalists, organization or participation in collective violence can produce material wealth that can enhance fitness [65, 73] as well as be passed down inter-generational channels [65, 74]. The key distinction is that foragers receive high payoffs for direct behavioral competition while farmers (and other delayed-return societies) benefit from resource-based competition. Seeming exceptions prove the rule: in prehistoric California, delayed-return hunter-gatherers who rely on the storage of privatized resources follow the violence pattern observed here among agriculturalists [6].

The positive covariance between violence and environmental productivity among foragers and horticulturalists speaks to the capacity of rich environments to finance frequent costly displays and render other forms of signaling less effective. For example, when high-ranking prey items are abundant and predictable, they may require less knowledge and skill to capture relative to environments where similar game is scarce and difficult to acquire [75]. In this case, the same strategy (e.g., big game hunting) will produce larger signaling payoffs in the poor environment relative to the rich one. Such ecological constraints on the number of honest signals that can be conveyed may result in a greater emphasis on status-driven violence, where abundant resources permit greater energy being allocated into violent competition. More frequent displays in productive environments may intensify competition, with positive feedback resulting in higher rates of violence. The negative covariance between violence and standard deviation in environmental productivity suggests that homogeneous environments promote violence either because abundant resources are spread evenly across the landscape, permitting a greater number of individuals to pay the steep costs of participation, or because other venues for status competition are rendered less effective when resources are evenly distributed.

Contrasting an immediate-return economy such as mobile foraging to a delayed-return economy like farming, the latter provides greater opportunities for resources to be stored, owned, and accumulated as wealth, which can be used to facilitate marriages or enhance status via conspicuous displays or generosity [12]. Following marginal utility theory, resource-related violence should be especially intense when resources are scarce, and stored in large, dense, monopolizable packages [6, 28–30]. This is because the utility of a resource diminishes with the amount of that resource an individual possesses [76] and risk preferences vary as a function of relative wealth [77, 78]. As a result, farmers (and likely pastoralists) with abundant resources should be more tolerant of theft and more risk-averse (in this case, violence-avoiding), whereas those with scarce resources should be less tolerant of theft and more risk-prone (violence-prone) [28]. To summarize, among delayed-return economies low productivity, high variance environments should promote violence from those seeking resources that can be translated to fitness payoffs and those incentivized to defend them [6, 12, 29].

## Violence and starvation

One might argue that the link between resource scarce environments and violence may indicate starvation-induced behavior. Afterall, if cooperative solutions to acute starvation are unavailable, individuals will use zero-sum strategies (including violence) to obtain

necessary subsistence resources regardless of MoS or other intervening variables such as age or sex. Nonetheless, we expect starvation-induced violence to be rare for three reasons.

First, if survivorship is frequently in jeopardy, one should see far broader demographic participation in resource conflict. Afterall, if starvation negatively impacts everyone, resource-based violence should involve a wide demographic subset of the population and far greater sex-based parity. Instead, intergroup violence most often involves the organization and participation of young adult males [28, 79, 80]. This is not to say that exceptions do not exist, but rather to highlight the general pattern.

Second, many of those who attempt to empirically tie violence to acute starvation or chronic nutrient deprivation have failed to do so effectively [e.g., 81], with violence more consistently resulting in *increased* food shortages [11, 34].

Third, as stated above, the 'resource procurement' hypothesis implies the capture or defense of group-level rewards, such as clustered resources or territory, that elicit collective action problems [25, 32]. Recent studies have suggested that collective action problems can be overcome when participation in inter-group violence serves to accumulate resource wealth that can be translated into fitness rewards, such as marriage opportunities [12] or status [11], suggesting resource procurement may be a proximate means of gaining social rewards rather than alleviating resource shortages.

This is not to say that human populations never undergo starvation, or that violence and starvation are totally unrelated. Rather, these factors indicate that starvation induced violence cannot explain the full, or even normal, range of variability in violence, and is thus not the primary target of selection. Instead, we argue that resource-based violence is primarily a strategy for obtaining material resources that can increase fitness by enhancing status and facilitating marriages and alliances.

## Conclusion

We find that violence-promoting environmental conditions are opposite for foragers and farmers, with violence among foragers peaking in rich, homogeneous environments, and violence among agriculturalists peaking in marginal, heterogeneous environments. We argue that the opposite covariance reflects the increasing role of material resources in motivating violent competition over fitness-related payoffs such as status, marriage opportunities, and alliances. These results yield insight into the behavioral ecology of violent behavior by assessing how variability in ecological conditions and subsistence economies explain cross-cultural differences in rates of violence.

The results of this paper should, of course, be treated as presenting tentative and testable hypotheses for future investigation. More robust tests of the expectations laid out in this paper will require a larger pool of archaeological trauma datasets that differentiate between ante-mortem and peri-mortem trauma, especially among mobile hunter-gatherers, horticulturalists, and pastoralists. Ethnographic datasets must be broadened to evaluate rates of violence among complex agricultural societies (particularly non-western populations).

The present study represents a preliminary effort to address the behavioral ecology of violence across the ecological and socioeconomic spectrum. Future theoretically-grounded research will benefit from a focus on explaining environmentally mediated variability in violent behavior, with an emphasis on individual motivations as a function of differential payoffs. The evolutionary perspective presented here can help to address questions regarding the frequency and intensity of human conflict, thus providing a foundation for developing science-based tools that may aid in conflict mitigation both in the present and the future.

## Supporting information

**S1 File. The complete dataset used for our analysis.**
(CSV)

**S2 File. R markdown of our complete analysis.**
(PDF)

## Acknowledgments

The authors thank the University of Utah Archaeological Center, the PLoS ONE editors, and the anonymous reviewers.

## Author Contributions

**Conceptualization:** Weston C. McCool, Kenneth B. Vernon, Peter M. Yaworsky, Brian F. Codding.

**Data curation:** Weston C. McCool, Kenneth B. Vernon, Peter M. Yaworsky, Brian F. Codding.

**Formal analysis:** Weston C. McCool, Kenneth B. Vernon, Peter M. Yaworsky, Brian F. Codding.

**Investigation:** Weston C. McCool, Kenneth B. Vernon, Peter M. Yaworsky, Brian F. Codding.

**Methodology:** Weston C. McCool, Kenneth B. Vernon, Peter M. Yaworsky, Brian F. Codding.

**Project administration:** Weston C. McCool, Brian F. Codding.

**Resources:** Weston C. McCool.

**Supervision:** Weston C. McCool, Brian F. Codding.

**Validation:** Weston C. McCool, Kenneth B. Vernon, Peter M. Yaworsky, Brian F. Codding.

**Visualization:** Weston C. McCool, Kenneth B. Vernon, Brian F. Codding.

**Writing – original draft:** Weston C. McCool, Kenneth B. Vernon, Peter M. Yaworsky, Brian F. Codding.

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
