## [Decision Letter · Decision Letter 0]

22 Nov 2021

PONE-D-21-31553Subsistence strategy mediates ecological drivers of human violencePLOS ONE

Dear Dr. McCool,

Thank you for submitting your manuscript to PLOS ONE. After careful consideration, we feel that it has merit but does not fully meet PLOS ONE’s publication criteria as it currently stands. Therefore, we invite you to submit a revised version of the manuscript that addresses the points raised during the review process.

As you address the reviewers’ comments, we would like for you to attend in particular to: 

(1) Reviewer #1’s requests for clarification;

(2) the same reviewer’s concern that environmental change complicates the assumption that modern estimates of NPP accurately reflect prehistoric ones; 

(3) providing additional support for the assumption that skeletal trauma is a reliable measure of violence per se, a concern shared by both reviewers.

We look forward to receiving your revised manuscript.

Kind regards,

Raven Garvey

Academic Editor

PLOS ONE

3. We note that figure 1 in your submission contain map images which may be copyrighted. All PLOS content is published under the Creative Commons Attribution License (CC BY 4.0), which means that the manuscript, images, and Supporting Information files will be freely available online, and any third party is permitted to access, download, copy, distribute, and use these materials in any way, even commercially, with proper attribution. For these reasons, we cannot publish previously copyrighted maps or satellite images created using proprietary data, such as Google software (Google Maps, Street View, and Earth). For more information, see our copyright guidelines: http://journals.plos.org/plosone/s/licenses-and-copyright.

   1. You may seek permission from the original copyright holder of Figure(s) 1 to publish the content specifically under the CC BY 4.0 license. 

Reviewers' comments:

Reviewer's Responses to Questions

**Comments to the Author**

1. Is the manuscript technically sound, and do the data support the conclusions?

Reviewer #1: No

Reviewer #2: Partly

2. Has the statistical analysis been performed appropriately and rigorously? 

Reviewer #1: I Don't Know

Reviewer #2: No

3. Have the authors made all data underlying the findings in their manuscript fully available?

Reviewer #1: No

Reviewer #2: Yes

4. Is the manuscript presented in an intelligible fashion and written in standard English?

Reviewer #1: No

Reviewer #2: Yes

5. Review Comments to the Author

Reviewer #1: This paper is based on a profound misunderstanding of violence and warfare. This is manifest in the opening abstract which ends with the statement: “We suggest that as societies transition from immediate return economies (e.g., foragers) to delayed return economies (e.g., farmers) material resources become an increasingly important motivation for interpersonal violence and warfare.” Interpersonal violence and warfare are two very different cultural phenomenon and cannot be conflated. Wife-beating is interpersonal violence and it can be followed and explained in the archaeological and ethnographic records; however, wife-beating and organized, lethal warfare between armed groups have completely different causalities and trajectories. To lump together all skeletal trauma and labeling it “violence” fails to distinguish accidental trauma (breaking one’s leg from a fall) from the kind of systematic trauma arising out of warfare (forearm fractures, fontal bone fractures).

The authors also fail to take into consideration chronological variability in the occurrence of violence/warfare. They conflate centuries or even millennia into single events, without realizing that violence varies markedly over time in any given area. This fact, by itself, argues against their primary hypothesis that there is a strong causal correlation between violence/warfare and subsistence economy. People in one area, with a particular subsistence economy can experience a century of warfare, and many centuries without violence. There is no sense of chronological control in this study and no effort to look at when warfare occurs and when it fades away. To simply impute warfare to skeletal trauma for a particular world area represents a basic misunderstanding of the data.

In an attempt to push an incredibly complex and diverse global data base into their hypothetical models, they have papered over so much variability as to make their conclusions irrelevant. I cannot finish this review without commenting that is singularly the most jargon-laden, almost incomprehensible paper on this topic that I have encountered. Had the authors given more consideration into making their arguments coherent rather than technical, they might have identified some of their basic flaws.

Reviewer #2: Dear authors

this is an interesting manuscript that I enjoyed reading. I add several comments below, which I hope you will find useful. In my view, the two critical issues of your work is to add a stronger discussion over the potential bias introduced by combining together archaeological and ethnographic data, and and on the use of NPP to measure ecological effects on violence.

Kind regards

Abstract

Line 47: I don't think 'career' is a good word here; you could use 'behaviour' or 'social behaviour'

Introduction

This section is a bit too concise, which negatively affects clarity, and it leaves several questions unanswered. A lot of work done on violence in ancestral human societies has used data on different geographic areas and cultures, including work you cite (e.g. Bowles 2009), so it is not clear why and how your cross-cultural approach is novel. Also, theories on violence from behavioural ecology make predictions that can be tested and potentially proved to be correct, irrespective of the cultural milieu in which violence is observed. You also don't cite some seminal work on the topic, including on the role of culture, for example Zefferman & Mathew 2005.

Line 67: It would be useful to have a definition of violence at the start of the introduction. Do you use the term for all sorts of violent-like behaviours, including bullying and passive aggression, or it's only for potentially lethal violence?

Lines 80-82: I think you should mention here that some studies have not found a positive relationship between participation to war and adaptive benefits (e.g. Ferguson 1989; Beckerman et al 2009).

Lines 83-88: I don't fully get your point here. You say you want to focus on cross-cultural variation and not on the payoffs of violence, but then your aim is to analyse the ecological and economic conditions triggering violence. Ecological and economic conditions are clearly linked to payoffs, so I don't understand the distinction you are trying to make. Moreover, ecological and cultural factors are linked together; for example, a group living in a highly seasonal environment may be forced to fight with other groups during periods of food scarcity and also develop group norms that reward warriors, both factors affecting payoff. I think this section should be clarified to make your point stronger.

Line 107: it would be useful to explain in details how your dataset represent an expansion of previous ones. Is this in relation to societies included, precision of the data, range of variables considered or else?

Methods

Lines 132-134: you should discuss here whether/to what extent you can reliably say that these signs of trauma are due to pre-/peri-mortem violence or to post-mortem rituals. These has been a lot of debate whether traumas on bones are really reliable indicators of violence; e.g. see Fry's book (2013).

Line 136: you should give the definition of NPP here, the first time you introduce this acronym

Lines 136-138: if this is a recommendation for future studies, it would be better placed in the conclusions

Line 163: how far back in time are the archaeological data from? These has been a lot of environmental changes between prehistoric times and now, and if we also include global warming I don't think NPP data collected between 2000 and 2015 is a reliable measure of habitat productivity to be linked to violence occurring thousands (or even hundreds) of years before. You claim that "While NPP provides a measure of modern productivity, the relative NPP ranking of each society should have remained consistent over time due to their broad geographic range representing general physiographic regions rather than variants within single ecological or climatological regimes", but this sounds a weak claim to me. Do you have any evidence to support this claim? is there any reference to previous work that has tested this assumption? There have been both micro-and macro-climatic changes at the end of the Holocene, that have been suggested to lead to changes in social organisation and the emergence of agriculture (e.g. Shennan 2018), so this point is really critical for the reliability of your findings.

Line 181: it would be useful to add a binary control variable, at least in the preliminary analyses, to see if you find any difference between archaeological and ethnographic data, otherwise you cannot really disentangle whether difference between type of societies are real or due to the fact that different data are available for different societies

Line 191: I believe you should compare a null model (containing no predictor variables) with your two base and full models. If there is no significant improvement from the null model, you should make no claims that NPP affects violence.

Results

I don't understand from this section or from the methods, whether you have data on MoS for the archaeological and ethnographic data or just for the latter. Why not analysing the two datasets separately, since you say above that there are differences in how they can estimate violence?

Discussion

Your results are in line with what one would predict based on previous work in behavioural ecology. However, I cannot see any discussion about the potential bias in the findings, introduced by combining together archaeological and ethnographic data, and on how reliable NPP is to test your predictions. These are key aspects that would need to be addressed or at least highlighted to the reader to help a more critical evaluation of your findings.

6. PLOS authors have the option to publish the peer review history of their article (what does this mean?). If published, this will include your full peer review and any attached files.

Reviewer #1: No

Reviewer #2: No

---

## [Author Response · Author response to Decision Letter 0]

6 Jan 2022

January 3, 2021 

PONE-D-21-31553

Subsistence strategy mediates ecological drivers of human violence

PLOS ONE

We would very much like to thank the reviewers for their incisive and valuable comments. 

Our revised manuscript and the responses presented here focus on addressing the three main concerns expressed by the academic editor. Below, we provide a point-by-point response to reviewer comments. Where major in-text changes were made we highlight revised text with red font. 

Academic editor’s main points: 

(1) Reviewer #1’s requests for clarification

(2) the same reviewer’s concern that environmental change complicates the assumption that modern estimates of NPP accurately reflect prehistoric ones.

(3) providing additional support for the assumption that skeletal trauma is a reliable measure of violence per se, a concern shared by both reviewers.

(4) The map figure copyright concern: the map used in our manuscript is an original creation by the authors and thus does not have any copyright violation.

Reviewer #1

Comment: Interpersonal violence and warfare are two very different cultural phenomenon and cannot be conflated. Wife-beating is interpersonal violence and it can be followed and explained in the archaeological and ethnographic records; however, wife-beating and organized, lethal warfare between armed groups have completely different causalities and trajectories. 

Response: Great point regarding definitions. We clarify throughout the revised manuscript to focus on interpersonal (primarily inter-group) violence. However, we should note that intra- and inter-group violence tend to co-occur (Lutz 2007; McCool et al. 2020; Nordstrom, 1998; Tung, 2014), and that changes in intra-group violence should be affected by the process outlined in our theoretical setup. For example, resource scarcity may promote intra-group competition that increases interpersonal violence (Daly, 2017), and status and mating competition may intensify within group animosities that result in interpersonal violence (Chagnon, 2013). 

Comment: To lump together all skeletal trauma and labeling it “violence” fails to distinguish accidental trauma (breaking one’s leg from a fall) from the kind of systematic trauma arising out of warfare (forearm fractures, fontal bone fractures).

Response: We apologize for the confusion. Our archaeological dataset relies entirely on publications that explicitly provide trauma data that is attributable to interpersonal violence and excludes accidental trauma and post-mortem damage. We have revised our methods section to clarify. 

Comment: The authors also fail to take into consideration chronological variability in the occurrence of violence/warfare. They conflate centuries or even millennia into single events, without realizing that violence varies markedly over time in any given area. This fact, by itself, argues against their primary hypothesis that there is a strong causal correlation between violence/warfare and subsistence economy. People in one area, with a particular subsistence economy can experience a century of warfare, and many centuries without violence. There is no sense of chronological control in this study and no effort to look at when warfare occurs and when it fades away. 

Response: We feel it is important to note that our primary result is not that there is “a strong causal correlation between violence/warfare and subsistence economy” (nor is it our “primary hypothesis”). It is the case with all multiregional archaeological analyses that chronological gaps will exist. Afterall, there is not a single region in which complete temporal resolution exists for data on violent trauma. Given this, we flatten temporal depth because we are interested in whether large physiographic regions differ in average rates of violence in response to variation in macroscale ecological conditions. Far from unusual, averaging diachronic variability is standard practice for multiregional studies in bioarchaeology (e.g., Allen et al. 2016; Arkush and Tung 2013 (flatten intraperiod variation); Beier et al. 2020; Delgado-Darias 2017; Fibiger et al. 2013; Kohler et al. 2014; Milella et al. 2020; Mummert et al. 2011; Pomeroy and Stock 2012; Scott and Buckley 2010:512; Standen et al. 2020; Steadman 2008:59; Vercellotti et al. 2014; Zhang et al. 2020), particularly in areas with limited radiocarbon datasets. 

Comment: To simply impute warfare to skeletal trauma for a particular world area represents a basic misunderstanding of the data.

Response: As stated in the methods section, the archaeological dataset is a measure of generalized interpersonal violence, not warfare. Our revised methods section also points out that our dataset does not include all skeletal trauma, only violent trauma. Because rates of intra-group violence tend to be low in small-scale societies (Bohem 1999; Wrangham et al. 2006) and co-occur with inter-group violence (Nordstrom 1998; McCool et al. 2020; Lutz 2007; Tung 2014) – see revised intro and methods sections – changes in rates of population-level generalized violence will largely be attributable to fluctuations in rates of inter-group violence, which should co-occur with intra-group conflict. However, as stated above, changes in intra-group violence should be affected by the process outlined in our theoretical setup. Our main goal with this paper is to assess how macroscale ecological conditions impact generalized rates of interpersonal violence, and whether subsistence economy has a mediating effect. 

Comment: In an attempt to push an incredibly complex and diverse global data base into their hypothetical models, they have papered over so much variability as to make their conclusions irrelevant. I cannot finish this review without commenting that is singularly the most jargon-laden, almost incomprehensible paper on this topic that I have encountered. Had the authors given more consideration into making their arguments coherent rather than technical, they might have identified some of their basic flaws.

Response: We contend that a key strength of our dataset is that it is “complex and diverse”. It is not possible to address the reviewer's concern that “they have papered over so much variability as to make their conclusions irrelevant” without additional details regarding how the complexity and diversity of our dataset renders our model results (from which our conclusions follow) “irrelevant.” As for the comment regarding the language and clarity of our paper, it is fully in-line with the broader literature on the evolutionary and ecological drivers of human violence (e.g., Allen et al. 2016; Bowles 2009; Glowacki et al. 2017; Zefferman and Mathew 2015; etc.). We suggest that the “incomprehensibility” may stem more from divergent research traditions between the authors and reviewer, rather than incoherent writing. 

Reviewer #2

Comment: Line 47: I don't think 'career' is a good word here; you could use 'behaviour' or 'social behaviour'

Response: Changed to ‘behavior.’

Comment: This section is a bit too concise, which negatively affects clarity, and it leaves several questions unanswered. A lot of work done on violence in ancestral human societies has used data on different geographic areas and cultures, including work you cite (e.g. Bowles 2009), so it is not clear why and how your cross-cultural approach is novel.

Response: We have expanded the introduction, included new citations, and worked to clarify our setup with more explicit theoretical expectations. While our primary focus is on the adaptive payoffs of interpersonal and inter-group violence, our revised introduction includes content recognizing some of the broader research efforts, including: “Recent research also explores the coevolution of inter-group violence with intra-group cooperation, altruism, and the emergence of complex social systems (Bowles 2009; Rusch 2014; Zefferman and Mathew 2015).” 

 We agree that a lot of excellent work has been done on cross-cultural approaches to human violence. However, the majority of the formal cross-cultural research focuses on (1) the co-evolution of coalitional violence with cooperation, parochial altruism, and cultural group selection (e.g., Bowles 2009), (2) the categories of available rewards for participation (e.g., Glowacki and Wrangham 2013), and (3) punishing free-riders (e.g., Mathew and Boyd 2011). While these studies are vitally important, our study provides a novel perspective by seeking to explain how variation in large-scale ecological and economic conditions structure the social and material payoffs for inter-group (and interpersonal) violence. To our knowledge, there is no published research evaluating how macroscale resource conditions and modes of subsistence affect the adaptive payoffs for violence, especially on the scale presented in our research. Our findings: that violence among foragers and horticulturalists peak in productive, homogeneous environments, while violence among farmers peaks in marginal, heterogeneous environments, are novel and have implications for the evolution of interpersonal violence and warfare. 

Comment: You also don't cite some seminal work on the topic, including on the role of culture, for example Zefferman & Mathew 2005.

Response: Thank you for clueing us into this literature. This reference and several like it have been included in our revised introduction. 

Comment: Line 67: It would be useful to have a definition of violence at the start of the introduction. Do you use the term for all sorts of violent-like behaviours, including bullying and passive aggression, or it's only for potentially lethal violence?

Response: We clarify throughout to focus on inter-group violence (warfare, raids, feuds) and to a lesser extent general interpersonal violence, although much of the processes outlined in the paper should impact both intra- and inter-group violence (e.g., status and mating competition). 

Comment: Lines 80-82: I think you should mention here that some studies have not found a positive relationship between participation to war and adaptive benefits (e.g. Ferguson 1989; Beckerman et al 2009).

Response: We include Beckerman et al. 2009 in the revised introduction to show that debate persists. We do not include the Ferguson 1989 paper as Chagnon 1989 provides a robust refutation of Ferguson’s claims. 

Comment: Lines 83-88: I don't fully get your point here. You say you want to focus on cross-cultural variation and not on the payoffs of violence, but then your aim is to analyse the ecological and economic conditions triggering violence. Ecological and economic conditions are clearly linked to payoffs, so I don't understand the distinction you are trying to make. Moreover, ecological and cultural factors are linked together; for example, a group living in a highly seasonal environment may be forced to fight with other groups during periods of food scarcity and also develop group norms that reward warriors, both factors affecting payoff. I think this section should be clarified to make your point stronger.

Response: We have expanded our theoretical setup to provide greater clarity and set out more explicit expectations. As suggested by the reviewer, resource violence and social rewards may very well be linked, and we address this possibility in the revised introduction and discussion sections. However, many evolutionary anthropologists studying violence find resource procurement to be an irrelevant or epiphenomenal motivation (e.g., Chagnon 2013; Glowacki and Wrangham 2013; Kelly 2005; Macfarlan et al. 2018), instead positing that direct access to social rewards (i.e., status, mating, alliances) provide the key indirect fitness payoffs. Alternatively, many scholars studying inter-group violence propose that the primary motivation for violence is the procurement or defense of scarce, monopolizable resources (e.g., Allen et al. 2016; Ember and Ember, 1992; Field, 2008; LeBlanc 1999; Vayda, 1976), with social rewards taking an epiphenomenal position. 

 We propose to test these hypotheses by noting that each hypothesis implies a different functional relationship between violent behavior and socioecological conditions. If the resource procurement hypothesis is correct (if the payoffs to violence are primarily material resources), then violence should positively co-vary with a scarce and variable resource distribution. If the social rewards hypothesis is correct (if the payoffs to violence are social rewards), then violence should positively covary with an abundant and predictable resource distribution. We recognize in the revised introduction that, as the reviewer states, fighting over resources can be linked to social rewards, but contend that in this case resource violence would only be initiated with resource scarcity, and would diminish as resources become increasingly abundant and predictable. In addition to this basic test, our analysis offers a novel avenue for the assessment of resource violence, in particular whether it yields increasingly high payoffs with the emergence of sedentism, storage, privatization, and wealth. We summarize these divergent expectations with a new figure (Fig 1). 

Comment: Line 107: it would be useful to explain in details how your dataset represent an expansion of previous ones. Is this in relation to societies included, precision of the data, range of variables considered or else?

Response: We added: “The dataset developed here builds off prior studies by providing additional archaeological and ethnographic samples and improves precision by excluding archaeological groups with small sample sizes.” 

Comment: Lines 132-134: you should discuss here whether/to what extent you can reliably say that these signs of trauma are due to pre-/peri-mortem violence or to post-mortem rituals. There has been a lot of debate whether traumas on bones are really reliable indicators of violence; e.g. see Fry's book (2013).

Response: Thank you for this suggestion. The revised methods section states that bioarchaeologists have robust methods for differentiating these types of trauma. For each of our archaeological samples only violent skeletal trauma is included, with post-mortem damage/rituals and accidents being excluded. 

Comment: Line 136: you should give the definition of NPP here, the first time you introduce this acronym.

Response: This has been fixed. 

Comment: Lines 136-138: if this is a recommendation for future studies, it would be better placed in the conclusions

Response: We include this caveat in the methods section as a preemptive response to readers who may be wondering why our samples are clustered in certain world regions. This clustering is due to the paucity of existing data rather than our own oversight. We include an additional sentence in the revised conclusion section calling for additional data in more diverse ecological settings. 

Comment: Line 163: how far back in time are the archaeological data from? These has been a lot of environmental changes between prehistoric times and now, and if we also include global warming I don't think NPP data collected between 2000 and 2015 is a reliable measure of habitat productivity to be linked to violence occurring thousands (or even hundreds) of years before. You claim that "While NPP provides a measure of modern productivity, the relative NPP ranking of each society should have remained consistent over time due to their broad geographic range representing general physiographic regions rather than variants within single ecological or climatological regimes", but this sounds a weak claim to me. Do you have any evidence to support this claim? is there any reference to previous work that has tested this assumption? There have been both micro-and macro-climatic changes at the end of the Holocene, that have been suggested to lead to changes in social organisation and the emergence of agriculture (e.g. Shennan 2018), so this point is really critical for the reliability of your findings.

Response: To address this point, we added the following content to the revised methods section: “Modern NPP has been used to predict numerous prehistoric phenomena including population density (Bradshaw et al. 2019; Eriksson et al. 2012; Timmermann and Friedrich 2016), habitat colonization (Codding and Jones 2013), resource scarcity (Allen et al. 2016) and more. While NPP provides a measure of modern productivity, the relative NPP ranking of each region should have remained consistent over time due to their broad geographic range representing general physiographic regions (i.e., bioregions) rather than variants within single ecosystems (see, Bioregions 2020). To better illustrate this point, we take as an example two regions included in our sample: the North American Colorado Plateau (CP) and the Illinois river valley (IRV). While the absolute NPP of these two regions certainly fluctuated throughout the centuries, there was no point during which the “mountain grassland and scrublands” of the CP were more environmentally productive (higher NPP) than the “tropical and subtropical dry broadleaf forests” of the IRV (Bioregions 2020; Dinerstein et al. 2017). The relative differences in environmental productivity between these two regions are due to their differing physiographic characteristics resulting in very different ecologies, which have remained comparatively consistent for millennia. The same can be said about the relative variation in NPP between the other bioregions compared in our analysis and the observations clustered therein (e.g., the lowland neotropics of the Amazon Basin versus the Canadian boreal forests). At most, changes in absolute NPP would only serve to exaggerate or flatten the modeled curves, but would not lead to trend reversals.” 

Comment: Line 181: it would be useful to add a binary control variable, at least in the preliminary analyses, to see if you find any difference between archaeological and ethnographic data, otherwise you cannot really disentangle whether difference between type of societies are real or due to the fact that different data are available for different societies

Response: We include a binomial model in the supplement that evaluates data type. We also include a new “Data Compatibility” section in the revised methods that works to address this concern and others mentioned below. In the revised introduction and methods section we discuss the need to used mixed archaeology/ethnography dataset due to inherent limitations in the available violence data. 

Comment: Line 191: I believe you should compare a null model (containing no predictor variables) with your two base and full models. If there is no significant improvement from the null model, you should make no claims that NPP affects violence.

Response: In our revised manuscript we generate a null model and conduct a likelihood ratio test comparing the null model to the base model and the full model to the base model. The base model results in a significant improvement from the null model (X2 = 0.47, p = 0.0458), while the MoS (full) model results in a significant improvement from the base model (X2 = 2.13, p < 0.0001). More details are available in the revised manuscript and Supplement 2. 

Comment: I don't understand from this section or from the methods, whether you have data on MoS for the archaeological and ethnographic data or just for the latter. Why not analysing the two datasets separately, since you say above that there are differences in how they can estimate violence?

Response: We apologize for the confusion. We categorized all samples by MoS and include an additional sentence at the beginning of the revised methods section to clarify. We do not analyze the datasets separately because, due to differential research traditions, ethnographic data is restricted to forager and horticultural groups while archaeological data almost entirely focus on agricultural groups. Thus, the ethnographic dataset would not have sufficient sample size to test the agricultural MoS while the archaeology dataset would have the same problem for foragers and horticulturalists. Thus, the combined approach taken here is necessary until ethnographic violence data is obtained for agriculturalists (especially non-western, non-state societies), and archaeological violent trauma data is generated for forager and horticultural societies. 

Comment: Your results are in line with what one would predict based on previous work in behavioural ecology. However, I cannot see any discussion about the potential bias in the findings, introduced by combining together archaeological and ethnographic data, and on how reliable NPP is to test your predictions. These are key aspects that would need to be addressed or at least highlighted to the reader to help a more critical evaluation of your findings.

Response: We included additional content in the revised discussion and conclusion sections regarding the combining of archaeological and ethnographic datasets. We acknowledge that differing data types may affect our models. However, until better datasets are available (see revised methods section), the approach taken here constitutes the best practice available, and our results follow strongly from theoretical expectations. We expanded our conclusion section to include these caveats and how we propose to treat them. We also included additional content in the methods about the use of relative NPP values and physiographic characteristics.

---

## [Decision Letter · Decision Letter 1]

24 Feb 2022

PONE-D-21-31553R1Subsistence strategy mediates ecological drivers of human violencePLOS ONE

Dear Dr. McCool,

Thank you for submitting your revised manuscript to PLOS ONE. We appreciate your careful attention to the feedback provided on your original submission and feel that, with minor revisions, your manuscript can meet PLOS ONE’s publication criteria. Therefore, we invite you to submit a revised version of the manuscript that addresses the points raised during the review process.

As you attend to the reviewers' comments, we would like for you to attend in particular to  (1) Reviewer #1's requests for additional clarification and (2) Reviewer #2's comment related to lines 159 & 187.==============================

We look forward to receiving your revised manuscript.

Kind regards,

Raven Garvey, Ph.D.

Academic Editor

PLOS ONE

Journal Requirements:

Reviewers' comments:

Reviewer's Responses to Questions

**Comments to the Author**

1. If the authors have adequately addressed your comments raised in a previous round of review and you feel that this manuscript is now acceptable for publication, you may indicate that here to bypass the “Comments to the Author” section, enter your conflict of interest statement in the “Confidential to Editor” section, and submit your "Accept" recommendation.

Reviewer #2: All comments have been addressed

Reviewer #3: (No Response)

2. Is the manuscript technically sound, and do the data support the conclusions?

Reviewer #2: Yes

Reviewer #3: Yes

3. Has the statistical analysis been performed appropriately and rigorously? 

Reviewer #2: Yes

Reviewer #3: Yes

4. Have the authors made all data underlying the findings in their manuscript fully available?

Reviewer #2: Yes

Reviewer #3: Yes

5. Is the manuscript presented in an intelligible fashion and written in standard English?

Reviewer #2: Yes

Reviewer #3: Yes

6. Review Comments to the Author

Reviewer #2: Dear authors

I think you have addressed all my comments and the manuscript is clearer and more complete.

I suggest a few sections where additional revision would be useful:

Line 55, "We argue that the trend reversal between foragers and farmers": here I think farmers refer to agriculturalists, but at line 53 you use the term 'forager/farmers' for horticulturalists, so I am not sure where you draw the line between foragers and farmers in this sentence.

Lines 55-60: I am ok with your argument, but it doesn't clearly follow from your finding that resource availability affects differently foragers and farmers. For example, why would farmers acquire "material resources to be transformed into social payoffs" only or more in unproductive environments?

Line 73: I really don't think you can use data on chimpanzees to make a claim about all non-human animals. If you look at the literature on other species, you will find that lethal violence is probably way more common in humans in comparison to the majority of other primates and mammals.

Line 276: it should read "while FOR farmers VIOLENCE negatively co-vary with productivity and homogeneity"

Line 321 "directly elevate inclusive fitness." I think there should be some refs at the end of this sentence

Reviewer #3: This is an ambitious and interesting paper tackling a ‘Big Question’ regarding the environmental conditions that encourage interpersonal violence. While this is many ways it still a ‘work in progress’, given the enormous complexity of the undertaking, I feel that the paper’s publication even at this stage will spark useful debate and discussion.

As the authors have already addressed a previous round of comments, I have only a few suggestions for clarification on some points, but also a recommendation for what would be a more major undertaking (dividing the foraging sample, which might require a larger sample size), and so therefore it is not required but might be considered for future.

80/ Not sure that all would agree that those seeing inter-group conflict as rare/maladaptive have been effectively countered, though the authors do acknowledge that this debate continues.

Odd to rely on the ethnographic record alone for foragers and horticulturalists when there is considerable archaeological evidence that might be drawn upon, though it is admittedly patchy. Some of the previous studies cited do drawn upon the archaeological record for foragers.

159/ “This database operationalizes sigma by defining it as the number of individual skeletons with evidence of violent trauma divided by the total skeletal sample for each archaeological observation”; repeated line 187

Surely this defines prevalence rather than sigma (a measure of variability), though from ln 258 it is being used here as a measure of the ‘rate of violence’?

317/ “Ethnographers report evidence showing adult males frequently fight over mating opportunities and real or perceived infidelities.”

This seems to contradict the previous statement regarding the low incidence of in-group violence conflict, assuming that opportunities for such infidelities are likely to be mainly within the group

345/ “Seeming exceptions prove the rule: in prehistoric California, delayed-return hunter-gatherers who rely on the storage of privatized resources follow the violence pattern observed here among agriculturalists.”

So this is not taken into account in the analysis, and the ‘foraging’ mode of subsistence includes all h-g? Not clear how this follows the violence pattern seen among agriculturalists as stated, as h-g practicing storage are often in relatively rich environments (such as those of CA and the NWC), whereas the discussion here focussed on the increased violence among agriculturalists being predicted by more marginal/unpredictable environments. This seems a discrepancy rather than an accord.

Why not divide foragers? The division between immediate- and delayed-return h-g is well established and in many ways it is odder to lump them than to divide them.

382/ “First, if survivorship is frequently in jeopardy one should see far greater demographic parsimony regarding participation in resource conflict”

Not clear that this follows, as there is little point in all and sundry engaging in violence (children, the elderly?) when they are more likely to be a hindrance than help to those better suited to violent conflict through both age and skills (e.g., hunting).

7. PLOS authors have the option to publish the peer review history of their article (what does this mean?). If published, this will include your full peer review and any attached files.

Reviewer #2: No

Reviewer #3: No

---

## [Author Response · Author response to Decision Letter 1]

7 Apr 2022

We would like to thank the editor and reviewers for their insightful commentary. Below, we provide a point-by-point response to review comments. 

Reviewer comments:

Reviewer #2

Comment: Line 55, "We argue that the trend reversal between foragers and farmers": here I think farmers refer to agriculturalists, but at line 53 you use the term 'forager/farmers' for horticulturalists, so I am not sure where you draw the line between foragers and farmers in this sentence.

Response: We edited this sentence to refer specifically to agriculturalists. 

Comment: Lines 55-60: I am ok with your argument, but it doesn't clearly follow from your finding that resource availability affects differently foragers and farmers. For example, why would farmers acquire "material resources to be transformed into social payoffs" only or more in unproductive environments?

Response: Farmers should acquire fitness-linked resource in all environments, but where resources are scarce and unevenly distributed individuals will receive higher relative payoffs for using violence to gain resources. Those with abundant resources should be more tolerant of theft and less willing (relative to those in poor environments) to use risk-prone strategies such as violence to obtain resources. As such, we don’t argue that farmers use violence to obtain resources only in poor environments, but that the payoffs for violence will be higher relative to those living in productive, homogeneous environments. We provide more detail on these relative payoffs using marginal utility theory on lines 365 – 374.

Comment: Line 73: I really don't think you can use data on chimpanzees to make a claim about all non-human animals. If you look at the literature on other species, you will find that lethal violence is probably way more common in humans in comparison to the majority of other primates and mammals.

Response: We agree and find that this sentence is distracting and have thus deleted it to save this point of discussion for later in the manuscript. 

Comment: Line 276: it should read "while FOR farmers VIOLENCE negatively co-vary with productivity and homogeneity"

Response: This sentence has been fixed. 

Line 321 "directly elevate inclusive fitness." I think there should be some refs at the end of this sentence

Response: We have added four references that support this statement. 

Reviewer #3 

I have only a few suggestions for clarification on some points, but also a recommendation for what would be a more major undertaking (dividing the foraging sample, which might require a larger sample size), and so therefore it is not required but might be considered for future.

Comment: 80/ Not sure that all would agree that those seeing inter-group conflict as rare/maladaptive have been effectively countered, though the authors do acknowledge that this debate continues.

Response: Fair point. Our objective with this statement is to acknowledge that important debate continues, but that debate is not the focus of the present research. We added additional text to emphasize that the ‘effective counters’ are for most, but not all, cases. 

Comment: Odd to rely on the ethnographic record alone for foragers and horticulturalists when there is considerable archaeological evidence that might be drawn upon, though it is admittedly patchy. Some of the previous studies cited do drawn upon the archaeological record for foragers.

Response: We do include several archaeological samples for horticulturalists, but these data are indeed rare. As for foragers, most of the data, while extremely valuable, are highly localized and often suffer from exceedingly small samples sizes that are below our sample size cutoff (lines 163-164). We are in the process, over the next several years, of putting together a much larger archaeo-ethnographic database which will expand upon the present database and will hopefully provide a more comprehensive archaeological database on foragers. This does assume however that the recent increase in bioarchaeology studies on violence among small scale societies continues. For the time being however, we must rely on the limited information available and look forward to the production and release of additional datasets. We hope with the publication of this manuscript to encourage future work generating trauma datasets. 

Comment: 159/ “This database operationalizes sigma by defining it as the number of individual skeletons with evidence of violent trauma divided by the total skeletal sample for each archaeological observation”; repeated line 187

Surely this defines prevalence rather than sigma (a measure of variability), though from ln 258 it is being used here as a measure of the ‘rate of violence’?

Response: We changed “sigma” to “PV” for “proportional violence” and changed “rate of violence” to “proportion of violence” throughout. 

Comment: 317/ “Ethnographers report evidence showing adult males frequently fight over mating opportunities and real or perceived infidelities.”

This seems to contradict the previous statement regarding the low incidence of in-group violence conflict, assuming that opportunities for such infidelities are likely to be mainly within the group

Response: We feel this is an important point to make, as intra-group hostilities do certainly occur, even if the rates are much lower relative to inter-group rates. Our point here is that when mating rewards are at stake, motives for inter-personal violence should be consistent regardless of whether antagonists are within group or outsiders. 

Comment: 345/ “Seeming exceptions prove the rule: in prehistoric California, delayed-return hunter-gatherers who rely on the storage of privatized resources follow the violence pattern observed here among agriculturalists.”

So this is not taken into account in the analysis, and the ‘foraging’ mode of subsistence includes all h-g? Not clear how this follows the violence pattern seen among agriculturalists as stated, as h-g practicing storage are often in relatively rich environments (such as those of CA and the NWC), whereas the discussion here focussed on the increased violence among agriculturalists being predicted by more marginal/unpredictable environments. This seems a discrepancy rather than an accord.

Response: This an excellent point about CA and NWC in relation to environmental productivity and conflict. Unfortunately, we cannot further investigate empirically how these two areas fit into our models for two reasons: First, the NWC does not contain sufficient violent trauma data to be included in our models; Second, the Central California bioarchaeological database does not contain a measure for generalized violent trauma (which is how violence is measured for all of our archaeology samples), as the database divides all individual traumas into either blunt or sharp force. As such, individuals could potentially be double counted if they exhibit both sharp and blunt force trauma. We cite the CA study because these largely delayed-return foragers follow a pattern similar to the farmers in our model. While the Central CA coast is indeed productive, we would need to model this relationship to assess where they fit in along the NPP and violence spectra, which we cannot, unfortunately, do. 

Why not divide foragers? The division between immediate- and delayed-return h-g is well established and in many ways it is odder to lump them than to divide them.

Response: We completely agree that this analysis could be improved by converting categorial subsistence economies to a continuous scale from immediate- to delayed-return. However, we should note that the foraging societies included in our database are immediate-return, and have revised the text to include this information (line 127). For the present analysis, the pool of available data is simply not large enough to further divide these data into more fine-grained economic categories, especially in relation to model sample sizes. For now, we are specifying the best possible models with the given data, which we hope will improve in size in quality in the coming years. We hope our research here will encourage future research on prehistoric violent trauma, particularly among foragers and pastoralists. 

Comment: 382/ “First, if survivorship is frequently in jeopardy one should see far greater demographic parsimony regarding participation in resource conflict”

Not clear that this follows, as there is little point in all and sundry engaging in violence (children, the elderly?) when they are more likely to be a hindrance than help to those better suited to violent conflict through both age and skills (e.g., hunting).

Response: Fair point. We do not mean to suggest that demographic participation should be universal if violence functioned to prevent starvation, but merely to state that participation should be much broader than the typical pattern of young adult males. Certainly, if basic survivorship were in jeopardy and violence was the only means of somatic maintenance we should expect far greater sex-based parity in participation, yet this does not commonly occur. The point we make is simply that the common pattern of young adult male organization and participation in collective violence heavily implies motives other than basic somatic maintenance in the face of starvation. We added a statement to clarify that a key point is the notable absence of sex-based parity in participation.

---

## [Editor Report · Decision Letter 2]

26 Apr 2022

Subsistence strategy mediates ecological drivers of human violence

PONE-D-21-31553R2

Dear Dr. McCool,

We’re pleased to inform you that your manuscript has been judged scientifically suitable for publication and will be formally accepted for publication once it meets all outstanding technical requirements.

Within one week, you’ll receive an e-mail detailing any required amendments. When these have been addressed, you’ll receive a formal acceptance letter and your manuscript will be scheduled for publication.

Kind regards,

Raven Garvey, Ph.D.

Academic Editor

PLOS ONE
---

## [Editor Report · Acceptance letter]

13 May 2022

PONE-D-21-31553R2 

Subsistence strategy mediates ecological drivers of human violence 

Dear Dr. McCool:

I'm pleased to inform you that your manuscript has been deemed suitable for publication in PLOS ONE. Congratulations! Your manuscript is now with our production department. 

Kind regards, 

on behalf of

Dr Raven Garvey 

Academic Editor

PLOS ONE